# Assessing Adherence, Competence and Differentiation in a Stepped-Wedge Randomised Clinical Trial of a Complex Behaviour Change Intervention

**DOI:** 10.3390/nu12082332

**Published:** 2020-08-04

**Authors:** Alison Kate Beck, Amanda L. Baker, Gregory Carter, Chris Wratten, Judith Bauer, Luke Wolfenden, Kristen McCarter, Ben Britton

**Affiliations:** 1School of Medicine and Public Health, The University of Newcastle, Callaghan, NSW 2308, Australia; Amanda.Baker@newcastle.edu.au (A.L.B.); gregory.carter@newcastle.edu.au (G.C.); luke.wolfenden@health.nsw.gov.au (L.W.); kristen.mccarter@newcastle.edu.au (K.M.); 2Centre for Brain and Mental Health Research, University of Newcastle, Callaghan, NSW 2308, Australia; 3Department of Radiation Oncology, Calvary Mater Newcastle Hospital, Waratah, NSW 2298, Australia; chris.wratten@calvarymater.org.au; 4Centre for Dietetics Research, The University of Queensland, St Lucia, QLD 4072, Australia; j.bauer1@uq.edu.au; 5Hunter New England Health, New Lambton 2305, Australia; Ben.Britton@health.nsw.gov.au

**Keywords:** head and neck cancer, fidelity, behavioural medicine, translational research, motivational interviewing, behaviour change counselling

## Abstract

Background: A key challenge in behavioural medicine is developing interventions that can be delivered adequately (i.e., with fidelity) within real-world consultations. Accordingly, clinical trials should (but tend not to) report what is actually delivered (adherence), how well (competence) and the distinction between intervention and comparator conditions (differentiation). Purpose: To address this important clinical and research priority, we apply best practice guidelines to evaluate fidelity within a real-world, stepped-wedge evaluation of “EAT: Eating As Treatment”, a new dietitian delivered health behaviour change intervention designed to reduce malnutrition in head and neck cancer (HNC) patients undergoing radiotherapy. Methods: Dietitians (*n* = 18) from five Australian hospitals delivered a period of routine care and following a randomly determined order each site received training and began delivering the EAT Intervention. A 20% random stratified sample of audio-recorded consultations (control *n* = 196; intervention *n* = 194) was coded by trained, independent, raters using a study specific checklist and the Behaviour Change Counselling Inventory. Intervention adherence and competence were examined relative to apriori benchmarks. Differentiation was examined by comparing control and intervention sessions (adherence, competence, non-specific factors, and dose), via multiple linear regression, logistic regression, or mixed-models. Results: Achievement of adherence benchmarks varied. The majority of sessions attained competence. Post-training consultations were clearly distinct from routine care regarding motivational and behavioural, but not generic, skills. Conclusions: Although what level of fidelity is “good enough” remains an important research question, findings support the real-world feasibility of integrating EAT into dietetic consultations with HNC patients and provide a foundation for interpreting treatment effects.

## 1. Introduction

Patients with head and neck cancer (HNC) often struggle to maintain adequate nutrition throughout radiotherapy. Malnutrition during radiotherapy undermines a range of treatment outcomes, including ultimate survival [1]. Dietetic intervention can improve clinical outcomes [2]. However, engagement of HNC patients with dietetic intervention is complicated by a range of factors that hinder oral intake, such as local tumour effects, radiotherapy side effects (e.g., mucositis, dysgeusia, xerostomia, and fatigue [3]), and mental health related concerns (e.g., depression [4]). In light of the deleterious impact of malnutrition, encouraging engagement in effective dietetic intervention despite these barriers is central to improving clinical outcomes for HNC patients. Psychological approaches to behaviour change may play a role.

Treatment fidelity is central to the development, conduct, evaluation [5], dissemination, and implementation [6] of complex behaviour change interventions. Although the science of fidelity continues to evolve and consensus has yet to be reached regarding its components [7], key elements include adherence, competence, and differentiation [8,9]. Adherence refers to whether an intervention is delivered “as intended”, while competence refers to the skill with which an intervention is delivered [8,9]. Differentiation is defined as whether and how an intervention is distinguishable from comparison conditions, particularly with regard to the potential “active” elements of the intervention and relevant “non-specific” or “common” (e.g., therapeutic alliance) factors [8,9]. Failure to report on treatment fidelity means that what clinicians’ actually deliver and how well remains unknown. This limits conclusions regarding both the feasibility of delivering an intervention, and the degree to which treatment effect(s) and/or lack thereof can be attributed to the intervention under evaluation [10].

Unfortunately, despite the centrality of treatment fidelity to the design, evaluation, and reporting of complex interventions, there is considerable inconsistency in the application of fidelity concepts and strategies within the behaviour change literature. In a systematic review of over 300 health behaviour change interventions conducted between 1990 and 2000, more than half of the studies failed to report on treatment fidelity [8]. Furthermore, only 15.5% of studies met criteria for “high treatment fidelity”, which was defined to mean at least 80% adherence to the checklist recommendations [8]. Over a decade later, although improvements have been observed, adequate consideration of treatment fidelity is rare [11,12,13]. Inadequate consideration of treatment fidelity represents a key barrier to the successful implementation and dissemination of effective interventions [8].

### 1.1. Objectives and Importance

Improved assessment and reporting of treatment fidelity represent a key priority for behaviour change research [10,14]. The 2017 revision of the CONSORT Statement for randomised trials of non-pharmacologic treatments now includes an item on treatment fidelity [15]. Further, as outlined within Stage III of the National Institute of Health Stage Model for Behavioural Intervention Development [8] there is a need to ensure that promising interventions can be delivered adequately (i.e., with fidelity) by real world clinicians within real world settings [8]. Accordingly, improved attention to treatment fidelity has the potential to help bridge the widely acknowledged “science-to-service” gap [8]. Fidelity is therefore a particularly important consideration for translational research. 

In the current paper, we report treatment fidelity outcomes within a real-world multi-site stepped wedge randomised controlled trial of “EAT: Eating as Treatment”, a complex health behaviour change intervention designed to reduce malnutrition in HNC patients undergoing radiotherapy [16,17]. This intervention has been successfully piloted by a Health Psychologist [18]. It was then adapted for delivery by oncology dietitians and evaluated in a National Health and Medical Research Council funded multi-site stepped wedge randomised controlled trial (RCT) [16,17]. Patient outcomes from the stepped wedge RCT are reported separately [17]. For context, the primary outcome of nutritional status (as measured by the PG-SGA [19] was superior amongst intervention relative to control participants) β = −1.53; Confidence Interval = −2.93 to −0.13 [17].

### 1.2. The Aims of the Current Paper Are to

Estimate the adherence to the EAT Intervention, including the degree to which intervention components were delivered during dietetic consultations conducted after training in the EAT Intervention.Estimate the competence with which the EAT Intervention was delivered, including the degree to which sessions conducted after training met apriori competence benchmarks.Examine the differentiation of the intervention period from the control period by comparing adherence and competence outcomes, non-specific factors (therapeutic alliance) and treatment dose (session number and duration). We expected that intervention sessions would be differentiated from control sessions with regard to adherence and competence outcomes (i.e., increased delivery of the intervention and greater skills). No specific hypotheses were made regarding therapeutic alliance or dose, as it is unknown whether and/or how these factors may be influenced by training in the EAT Intervention.

This paper contributes to the growing body of research assessing treatment fidelity in complex behaviour change interventions. It also presents novel data, as it is the first published, large scale multi-site account of the application of multiple behaviour change strategies by dietitians within real world consultations with HNC patients.

## 2. Materials and Methods 

This study was conducted in accordance with the National Statement on Ethical conduct in Human Research [20]. Approval was granted by the Hunter New England Human Research Ethics Committee (HREC) of Hunter New England Health (HREC/12/HNE/108; Approval date 09 May 2012). Site specific HREC approvals were also granted.

### 2.1. Setting and Trial Design

A detailed description of study procedures [16] and fidelity methods [21] can be found in published protocols. Study sites comprised five major Australian hospitals located in South Australia, Victoria, Western Australia, and Queensland. Each hospital treated a minimum of 100 HNC patients per year and were recruited through the Trans-Tasman Radiation Oncology Group (TROG). A stepped-wedge RCT design was used to evaluate the impact of the EAT Intervention on the nutritional status of HNC patients undergoing radiotherapy. HNC patients were recruited in waves such that each cluster contributed data under both control and intervention conditions (See Appendix A). A stepped wedge-randomised design was used particularly to limit contamination across treatment condition [22], whilst reducing risk of bias by use of a randomised design. The order of movement to the intervention condition was randomised using a uniform random number generator in STATA. Dietitians were aware of the intervention condition. Participants and fidelity coders were blind to allocation.

### 2.2. Intervention Providers

A total of 24 radiotherapy dietitians (87.5% female) at five participating hospitals (located in Adelaide, Melbourne, Perth, and Brisbane) provided “treatment as usual” (TAU) to HNC patients enrolled in the EAT trial during the control period. Of these, 13 also participated in training and delivered the EAT intervention during the intervention period. Due to staff turnover, a further five dietitians (who only participated in the intervention period) were trained to deliver the EAT Intervention.

All intervention dietitians (*n* = 18) had attained Bachelor level qualifications. Nine had completed postgraduate level training in nutrition and dietetics (including masters or postgraduate diploma in nutrition/dietetics). Experience working with HNC patients ranged from a few months to more than 20 years (Mean = 3.38 years; Mode = 1 year). Detailed demographic information was not available for the dietitians who only saw patients in the control period (*n* = 11).

### 2.3. Interventions

#### Treatment as Usual

During the control period, dietitians were instructed to deliver TAU, making no changes to their routine delivery of dietetic care. Routine dietetic care of HNC patients, as defined by treatment guidelines [23], comprises dietary assessment, counselling, and intervention to prevent or minimise malnutrition. Recommended session frequency is once per week during radiotherapy, once per fortnight for the first six-weeks post-radiotherapy and “as needed” thereafter [23]. Within the sites involved in the current study, a “standard” consultation was scheduled for 20–30 min.

### 2.4. The EAT Intervention 

#### Intervention

A detailed account of the EAT Intervention is published [16,17,21]. The EAT Intervention is designed to support HNC patients to maintain adequate nutritional status despite a range of commonly encountered physiological (e.g., local tumour effects and treatment side-effects [3]) and psychological (e.g., depression [4]) barriers to adequate oral and/or enteral intake. Although EAT skills and principles are manualised, EAT is not a linearly structured intervention. Rather, it is based on motivational and behavioural principles (Figure 1) such that it can be flexibly integrated into all consultations. This is consistent with recommendations for balancing fidelity to the content and principles of an intervention with the pragmatic approach required when delivering an intervention within routine clinical practice [24]. It is informed by behaviour change counselling [25], an adaptation of motivational interviewing (MI), designed to maximise the effectiveness of behaviour change conversations held within time-limited consultations. MI is used during each consultation to elicit and reinforce patient reason(s) for maintaining adequate nutrition, performance of helpful nutrition related behaviours, and collaboratively identify relevant nutrition related goals (as needed). When goals are identified, these are simplified, documented, reviewed, and reinforced using a written nutrition planner. A standardised nutrition assessment is also performed during week one, and as needed thereafter to provide an objective measure of nutritional status across time.

The “EAT to LIVE” conversation (see Figure 1) draws upon established principles of behaviour change and requires the dietitian to (i) elicit patient motivation for undergoing radiotherapy treatment (i.e., survival); (ii) provide information and education to explicitly highlight the relationship between adequate nutritional status and radiotherapy treatment outcomes (i.e., remaining well-nourished increases likelihood of survival); (iii) explore the (in)consistency between current behaviour, nutritional status and what is needed to maintain an adequate nutritional status; and (iv) work towards developing a concrete plan of the behaviour(s) required to maintain and/or improve nutritional status. We expected this conversation to be held (at a minimum) during week five (when difficulties regarding nutritional intake typically peak). However, the EAT to LIVE conversation could be held again earlier and/or later according to dietitian judgement/clinical need (either in its entirety and/or component parts).

### 2.5. Procedure

#### 2.5.1. Training

Training in the principles, strategies, and timing of the EAT Intervention was delivered as part of an overall package designed to support systems level change [26]. Further details are provided in the published clinical trial [16] and fidelity [21] protocols. Training was designed to promote adherence to and competence in the delivery of the EAT Intervention via (i) information provision and discussion (regarding the background, rationale, principles, and strategies) and (ii) modelling, role play, self-reflection, and peer and trainer video feedback of intervention principles and strategies. Dietitians were trained to use principles of behaviour change (Figure 1) to guide all patient consultations. Training was provided by the same facilitators (trial Clinical/Health Psychologists authors AKB, BB and/or AB) and comprised an initial two-day workshop, followed by one-day clinical “shadowing”, whereby the trainers accompanied dietitians during usual practice to observe and provide “real-time” troubleshooting and feedback. Approximately two months after the initial workshop, trainers returned to the site to provide a one day “booster” workshop designed to support intervention adherence, and competence via practice and discussion of intervention principles and strategies. Trainers again accompanied dietitians during their routine consultations to further troubleshoot “real world” implementation of the EAT Intervention. Dietitians also attended regular (at least monthly) supervision and coaching with a trial Clinical Psychologist (AKB) throughout the duration of intervention delivery.

Training commenced March 2014 (i.e., when the first workshop was delivered to the first site) and concluded in September 2015 (i.e., when the last booster workshop was delivered to the last site). Due to staff turnover and/or extended leave, seven EAT trained dietitians (across three sites) left at some point during the trial. A condensed version of the workshop was delivered to five replacement dietitians (across two sites) and the study workload for two was distributed amongst the remaining trained dietitians at these two sites.

#### 2.5.2. Audio-Recordings

Consistent with the gold standard for evaluating the fidelity of intervention delivery [8,14,27] dietetic consultations were audio recorded throughout the duration of the trial (*n* = 303 study participants provided consent). A total of 1925 consultations were recorded from July 2013 until April 2016. This sample represents approximately 61% of the dietetic sessions identified via patient chart review, with a comparable number of control (63%) and intervention (60%) sessions recorded. Reasons for not recording sessions included clinician error, recorder malfunction, patient request, or unsuitability of the consultation location (e.g., telephone, inpatient, and/or common treatment area).

#### 2.5.3. Sampling

Approximately 20% of audio recordings were randomly selected for coding (control *n* = 196; intervention *n* = 194). Randomisation was conducted using the “random” function in Excel and stratified according to intervention period (control vs. intervention), hospital site (A vs. B vs. C vs. D) and when during the patient’s course of radiotherapy the dietetic consultation was held (week one of radiotherapy vs. week five of radiotherapy vs. all other weeks during radiotherapy vs. post-radiotherapy). Week one and week five of radiotherapy were chosen as strata based on apriori timing of intervention elements (i.e., “Nutrition Assessment” and “EAT to LIVE”, respectively). “During radiotherapy” and “post-radiotherapy” were chosen as strata to account for differences in side effects, nutritional needs, and the frequency of dietetic consultations across these treatment periods.

#### 2.5.4. Coding

Coding was completed by an independent assessor blind to the schedule of training and intervention content. Each session was listened to from start to finish, and coding was based on the entire session. Approximately 20% of coded tapes (*n* = 89) were randomly selected and (i) re-coded by the same independent assessor for intra-rater reliability and (ii) re-coded by a second independent assessor for inter-rater reliability (see Appendix A).

#### 2.5.5. Measures

##### Adherence

Adherence was assessed using a checklist developed by the research team and the Behaviour Change Counselling Index (BECCI; [28]). The study checklist (see Table 1) describes key intervention elements, with each item rated as being present (“yes”) or absent (“no”). Discussing patient reasons for undergoing radiotherapy (i.e., survival) and explicitly highlighting the relationship between (mal)nutrition and radiotherapy outcomes were expected to be unique features of the EAT Intervention. Conversely, nutrition assessment and dietary counselling (i.e., discussing the adequacy of nutritional intake) are recommended elements of routine care [23], and therefore also likely to occur during TAU. Anecdotal dietitian feedback suggested that although nutrition plans are also a recommended feature of routine care [23], in practice, they are rarely implemented.

The 11 item BECCI [28] was used to assess adherence to motivational interviewing. Each item is rated on a five point scale (0 = Not at all, 1 = Minimally, 2 = To some extent, 3 = A good deal, 4 = A great extent) and used to calculate an overall mean “practitioner score”. Published evidence lends support to the reliability and responsiveness of this instrument (Cronbach’s Alpha from α = 0.63 to α = 0.71; Inter-rater reliability from R = 0.79 to R = 0.93; Intra-rater reliability from R = 0.60 to R = 0.90; and Standardised Response Mean = 1.76; [28]).

To assess the degree to which dietitians embodied the “spirit” of intervention delivery (empathy, genuineness and warmth) we used the “interpersonal effectiveness” item from the Cognitive Therapy Scale–Revised (CTS-R; [29]). This item is rated on a scale of 0 to 6, with higher ratings indicating greater expression of warmth, concern, confidence, genuineness, and professionalism. The Dreyfus system can be applied to this item to produce a binary rating with “competent” levels of interpersonal effectiveness denoted by a score > 3 [29,30].

##### Competence

Due to the paucity of brief, validated tools available at study inception to assess competence in behaviour change counselling, the Cognitive Behaviour Therapy (CBT) competence item from the CTS-R; Reference [29] was modified to reflect competence in skills of behaviour change counselling. This item is rated on a scale of 0 to 6, with higher ratings indicating greater competence. “Competence” in behaviour change counselling is denoted by a score > 3 [29,30].

### 2.6. Non-specific Factors

“Non-specific” factors are those intervention elements that are shared or “common” to a range of interventions, irrespective of the underlying theoretical model. Therapeutic alliance was identified as an important non-specific factor due to the consistent and robust relationship demonstrated with client engagement [31] and effective intervention delivery [32] across a range of settings, patient groups, and intervention modalities [33,34]. Therapeutic alliance (patient and dietitian rated) was assessed using The Agnew Relationship Measure: Five Item Version (ARM-5 [35]). This instrument was included as part of the RCT outcomes assessment battery administered at each of the study assessment time points (first and last week of radiotherapy, four- and 12-weeks post-radiotherapy). Items are summed to produce an overall index of “core alliance”, with higher scores reflecting stronger therapeutic alliance.

### 2.7. Number and Duration of Dietetic Consultations

The number and date of each dietetic consultation attended throughout the trial was documented as part of a chart review conducted at the 12-week follow-up assessment for each study participant. Session duration (in minutes) was indexed by the length of the audio-recorded consultations. 

### 2.8. Statistical Analysis

Data analysis was conducted on all coded recordings (control *n* = 196; intervention *n* = 194) using the Statistical Package for the Social Sciences (SPSS), version 26.0.

#### 2.8.1. Adherence

The magnitude of dietitian use of the EAT Intervention was estimated by calculating (a) the proportion of recordings demonstrating each study checklist skill, with threshold for “high fidelity” set at 80% [14] and (b) BECCI practitioner score [36]. The benchmark for MI adherence was defined as a mean BECCI score of at least 2.57, based on the mean BECCI score achieved by dietitians during a pilot of training and fidelity methods [37]. The degree to which the intervention was delivered in the “spirit” of MI is estimated by reporting mean scores for the CTS-R interpersonal effectiveness item and calculating the proportion of tapes that achieved a score of at least three (i.e., competent; [30]).

#### 2.8.2. Competence

The magnitude of competence is estimated by reporting the mean score of the CTS-R “Application of Behaviour Change Counselling” item and the number of tapes that achieved a score of at least three (i.e., competent; [30]) on that item.

#### 2.8.3. Differentiation

To examine whether training in the EAT intervention produced differences in adherence, competence and dose, we entered intervention period (control vs. intervention) as a predictor into a series of multiple logistic regression analyses (dependent variables: study specific checklist items; BECCI adherence benchmark; and CTS-R application of behaviour change competence benchmark) or linear regression analyses (dependent variables: BECCI practitioner score, CTS-R application of behaviour change counselling item score, CTS-R interpersonal effectiveness item score, session number, and session duration) as appropriate. To control for clustering (hospital site A/B/C/D), when during the patient’s course of radiotherapy the dietetic consultation was held (week one/week five/all other weeks during RT/post RT) and background temporal effects that could have confounded the stepped-wedge design (days since the study commenced) these were also entered as covariates into all regression analyses.

#### 2.8.4. Non-specific Factors

Dietitian and patient self-reported therapeutic alliance scores were calculated according to published guidelines [35]. As therapeutic alliance was collected as part of the study assessment battery (i.e., relative to the 20% sample of audio-recorded consultations used to assess adherence and competence), the statistical analysis plan for the secondary outcomes of the EAT Trial [16] was employed. Therapeutic alliance was analysed using an intention to treat, Linear Mixed Models regression and conducted using STATA. The model utilised unstructured covariance and included the following: cluster level variables of intervention period (to evaluate the impact of training on therapeutic alliance); hospital (to adjust for differences between hospitals); individual level variables of baseline therapeutic alliance (to adjust for differences in baseline alliance); calendar time (to adjust for any background temporal effects that could have confounded the stepped-wedge design); and assessment moment (to adjust for differences in alliance through radiotherapy and recovery). The model included a random level individual level intercept to account for the repeated measures on individuals over assessment moments, and a random coefficient for assessment moment to allow for heterogeneity in subject specific trends.

An alpha level of 0.01 was used to signify a difference between intervention and control with regard to adherence, competence, non-specific factors (therapeutic alliance) or dose.

## 3. Results

### 3.1. Intervention Adherence

Post-training delivery of the EAT Intervention is presented in Figure 2. Within the week five intervention consultations sampled, of the two “unique” elements of the EAT Intervention “Eating as Integral”, but not “Reasons for Radiotherapy” met the threshold for “high fidelity”, being delivered in 88.2% and 61.7% of the week five sessions sampled, respectively. The development and review of a nutrition plan fell short of “high fidelity” delivery across all sessions sampled. Both “non-unique” elements were delivered with “high fidelity”, with “adequacy of intake” discussed in more than 97% of week five sessions and “nutrition assessment” conducted in more than 93% of week one consultations. All study skills were utilised in consultations conducted outside of week five of radiotherapy (i.e., “as needed”), and with the exception of “reasons for radiotherapy” continued to be integrated into post-radiotherapy consultations.

MI skills were applied “to some extent” in intervention sessions held in week one (M = 2.53, SD = 1.39), week five (M = 2.29, SD = 0.44), during radiotherapy (M = 2.14, SD = 0.42), and post-radiotherapy (M = 2.09, SD = 0.36). Almost 30% of week five consultations (29.4%), and fewer than 20% of all other consultations (6.3%, 16.7%, and 10% for week one, during radiotherapy and post-radiotherapy, respectively) attained a mean benchmark BECCI score of 2.57 (See Figure 2).

All intervention sessions sampled were conducted in the “spirit” of MI (empathy, genuineness, warmth, and collaboration), with no CTS-R Interpersonal Effectiveness Scores falling below three.

### 3.2. Intervention Competence

The mean score for the CTS-R behaviour change counselling competence item was in the “competent” range for sessions held in week five of radiotherapy (M = 3.08, SD = 1.42) and in the “advanced beginner” range for week one (M = 2.53, SD = 1.39), during radiotherapy (M = 2.73, SD = 1.38), and post-radiotherapy (M = 2.72, SD = 1.34). At least half of post-training sessions held in week one (50%), week five (58.8%), during (57.7%), and post (56%) radiotherapy met the threshold for competence (>3) in behaviour change counselling (see Figure 1).

### 3.3. Delivery of the EAT Intervention Across Time

Dietitian use of the EAT intervention throughout the intervention period was found to be consistent across time. Neither the delivery of, nor competence in The EAT Intervention was found to significantly differ as a function of time since training.

#### 3.3.1. Differentiation

Intervention adherence, competence, non-specific factors and dose during control and intervention period are compared in Table 2. In dietetic consultations conducted during the intervention period, a clear increase in dietitian use of skills consistent with the EAT Intervention is evident. Regression analyses confirmed that dietitians were more likely to utilise both “unique” intervention elements during consultations held after training, with exploration of patient reasons for undergoing radiotherapy increasing from largely absent, to almost a quarter of all sessions sampled (1 vs. 22.1%), while dietitian use of information and education to highlight the relationship between (mal)nutrition and radiotherapy outcomes doubled (25 vs. 50%). Intervention sessions were also significantly more likely to include a written nutrition planner (4 vs. 31.4%). Conversely, although a greater proportion of post-training consultations did include a collaborative review of a nutrition plan (1.2 vs. 18.5%) this difference did not reach significance after accounting for the effects of site, radiotherapy interval and calendar time. Similarly, for the two “non-unique” intervention features (i.e., those we expected to occur during routine dietetic consultations), neither the observed increase in the conduct of a nutritional assessment (65.3 vs. 80.9%) nor discussion of the adequacy of patient nutritional intake (88.2 vs. 96.3%) reached significance.

Post-training consultations were also characterised by greater use of behaviour change counselling. Regression analyses confirmed that the number of sessions meeting the BECCI benchmark for adherence to behaviour change counselling was in favour of post-training consultations (6.1 vs. 15.5%), and a significant, albeit modest increase was also observed for the overall BECCI Practitioner Score.

Dietitian competence in behaviour change counselling also favoured intervention sessions. Regression analyses confirmed that both the number of sessions attaining competence (35.2 vs. 56.2%) and the mean level of competence (2.21 vs. 2.72) achieved was significantly greater for sessions conducted after training in the EAT intervention.

The “spirit” with which consultations were delivered did not significantly differ between control and intervention consultations (95% CI = –0.246, 0.180; *p* = 0.762). Mean dietitian demonstration of empathy, warmth, genuineness, and concern during both control (M = 5.51, SD = 0.73) and intervention (M = 5.69, SD = 0.67) sessions fell within the “expert” range.

#### 3.3.2. Contamination

Thirteen audio-recorded consultations with control patients occurred after clinicians received training in the EAT Intervention. Both observations of “reasons for radiotherapy”, two occurrences of both “eating as integral” and “nutrition plan” and one of the observed times that a nutrition plan was reviewed occurred when control patients were seen by trained clinicians. “Adequacy of intake” was also discussed during 11 of these sessions and a nutrition assessment conducted in all thirteen.

### 3.4. Non-Specific Factors

Therapeutic alliance was comparable before and after training in the EAT Intervention (Table 2). Intention to treat linear mixed models regression confirmed that neither dietitian (95% CI = −7.28, 3.73; *p* = 0.528) nor patient (95% CI = −6.76, 3.55; *p* = 0.542) ratings of therapeutic alliance were significantly influenced by training in the EAT Intervention (Table 1).

### 3.5. Dose

The total number and mean duration of dietetic appointments did not significantly differ for control and intervention participants (Table 2).

## 4. Discussion

The integration of behaviour change techniques into the sessions sampled lends support to the feasibility of EAT intervention delivery under real world conditions. Dietitian use of and skill in the delivery of the EAT Intervention clearly favoured post-training consultations. The level of change varied according to the intervention skill assessed and few elements were delivered according to apriori benchmarks. However, post-training consultations were clearly distinct from routine care regarding dietitian use of motivational and behavioural strategies.

### 4.1. Delivery of Study Checklist Skills

During week five of radiotherapy when we expected the EAT to LIVE conversation to occur, the most marked change was in a central “unique” intervention element, the eliciting of patient motivation for undergoing radiotherapy (i.e., survival). This was absent during TAU, and present in more than 60% of week five sessions conducted after training. This is important since eliciting and reinforcing intrinsic client motivation is central to the role of MI in facilitating positive behaviour change [38]. In contrast, despite self-monitoring and structured meal plans being evidence based strategies in nutrition counselling [39,40], whose frequency improved after training, fewer than half of the week five sessions sampled demonstrated evidence of a written nutrition planner, and even fewer reviewed client progress toward previously developed plans. Surprisingly, empirical evidence for dietitian implementation of this evidence-based nutrition strategy is lacking. Given the importance of self-monitoring, action planning, and goal setting for promoting positive health behaviour change [41,42], further research is needed to understand the degree to which nutrition plans are used by oncology dietitians and how best to support their integration into routine dietetic care. Use of electronic systems to prompt clinicians [43] and/or electronic planning and monitoring via smart phone applications may help to overcome some of the challenges currently encountered with hardcopy written nutrition planners. Indeed, hand-held computers have been found to not only increase self-monitoring of nutrition goals, but also improve patient attitudes towards this important behavioural strategy [39].

During the intervention period all checklist items were present to varying degrees in sessions either side of week five of radiotherapy. This may be indicative of “flexibility within fidelity”, that is, tailored intervention delivery depending on client need [44], which is also consistent with the study protocol (i.e., utilising principles of behaviour change to guide the application of intervention component(s)). However, the contribution of clinician factors (e.g., preference and/or confidence) to the presence/absence of intervention components must also be acknowledged [45]. Including a measure of clinician attitude and/or preference (e.g., The Evidence-Based Practice Attitude Scale [46]) would lend insight into the potential relationship between clinician factors and fidelity of intervention delivery.

Some intervention elements were also present during sessions conducted before training. The two checklist items most frequently observed during control sessions (nutrition assessment and discussing the adequacy of the patient’s nutritional intake) were similarly employed in sessions conducted after training. Both skills are fundamental to nutritional counselling with HNC patients [23] and are therefore expected components of routine dietetic care. Conversely, the motivational and behavioural elements of the EAT Intervention were largely absent during treatment as usual. Given that therapeutic alliance, interpersonal effectiveness, and the number and duration of dietitian consultations remained stable, intervention sessions can be viewed as distinct from control sessions on the basis of the number and type of behaviour change principles and strategies employed. This finding provides an important foundation for interpreting treatment effects. Specifically, it improves our confidence in the likely contribution of the EAT intervention to the improved nutritional status demonstrated by intervention patients [17]. Analyses to explore the relationship between fidelity of intervention delivery and treatment outcome are underway and will be reported separately.

### 4.2. Behaviour Change Counselling

Modest improvements were also observed in dietitian use of and skill in behaviour change counselling. Post-training BECCI scores suggest that clinicians employed behaviour change counselling “to some extent”. This is greater than several published accounts, whereby post-training use of behaviour change counselling by medical students [47], primary care clinicians [48], and nurses [49] was “minimal”. Consistent with findings from a recent systematic review of training health care professionals in MI [50] dietitian competence also improved. Almost 60% of coded intervention sessions met threshold for “competence”, with an additional 22% demonstrating at least “beginner” levels of proficiency. This is also greater than the level of post-training proficiency in MI typically observed within the published literature [51]. These findings are likely a product of the comprehensive and intensive training package employed in the current study. The workshop afforded opportunities for skill and knowledge acquisition across the eight key stages for learning MI [52]. As a one-off workshop is insufficient to facilitate sustained clinician proficiency [53], dietitian behaviour change was then further supported via clinical shadowing, ongoing supervision, coaching, and feedback and booster training. Conversely, our findings also highlight the inherent challenge in training clinicians to consistently employ the collaborative, motivational, and behavioural principles embodied by this approach. Only modest improvements were observed despite employing a comprehensive training package comprising ongoing, evidence-based learning methods [54]. Given that “advising” [55] “instruction”, “teaching” and “lecture” [56] represent common methods utilised by dietitians (and healthcare providers more broadly), it is reasonable to suspect that consistent application of behaviour change counselling was undermined by dietitian reliance on these more familiar, less effortful modes of communication. Unfortunately, such didactic approaches to patient education and counselling are unlikely to promote health behaviour change [57] and may be experienced by patients as unhelpful, overwhelming, and discouraging [58]. Clearly, innovative approaches to improving dietitian use of effective, collaborative communication, and behaviour change techniques are needed. Indeed, dietitians have suggested that improved training and support in the application of behavioural and motivational approaches is needed [55,56]. Perhaps by ensuring that behaviour change theory, principles, and strategies are a central component of all training programmes from the outset, and conveyed using a combination of didactic and experiential learning. For example, ongoing role play, self-reflection, and feedback would provide trainee dietitians with experiential evidence for the relative utility of didactic vs. evidence based approaches to behaviour change, together with an opportunity to identify and overcome barriers; thereby, promoting competence and confidence in the use of evidence based approaches to behaviour change.

### 4.3. Implications for Research and Practice

Our findings clearly demonstrate a statistically significant improvement in the application of the EAT Intervention after training. Further research is needed to investigate whether it is possible to determine an empirically derived benchmark as to what level of fidelity is “good enough” [10]. As part of our practice change intervention [16,26] a benchmark of 80% (i.e., high fidelity; Reference [14] was selected. In the 20% sample of dietetic sessions rated, in all but three of the domains assessed, adherence fell short of this potentially ambitious benchmark. This is likely due, in part, to the translational nature of the current study. Indeed, relative to highly controlled efficacy and effectiveness studies, evidence suggests that adherence levels tend be lower when interventions are integrated by frontline workers into their usual consultations [59]. Further, drawing from the broader health behaviour change literature (e.g., smoking cessation and physical activity) it is not uncommon for fewer than 50% of behaviour change techniques to be delivered in practice [60].

This raises the question of whether an 80% benchmark is a reasonable adherence benchmark within the context of real-world consultations. This is especially pertinent for studies like ours, which adopt a manual guided, relative to manual driven approach. That is, clinicians are able to exercise their own professional judgement about how best to integrate the intervention into their work—an approach that has been linked to clinicians being more engaged, motivated, and effective [61]. Perhaps a more moderate benchmark is warranted, especially given that overly rigid (and overly lax) adherence to intervention delivery may undermine treatment outcomes [62]. However, what this benchmark should be remains to be determined.

Of particular importance to clinical practice, is that the observed increase in motivational and behavioural strategies employed during consultations conducted during the intervention period did not translate to an increase in session duration or session number. This highlights the feasibility of delivering EAT under real-world conditions, a key aim of Stage III research [8]. Furthermore, as time is commonly cited by dietitians (and health care clinicians more broadly [50]) as a key barrier to the application of behaviour change skills [63,64], future training iterations may therefore benefit from highlighting the time efficiency of the intervention and eliciting and addressing unhelpful expectations and predictions regarding ongoing application of the EAT Intervention.

### 4.4. Strengths and Limitations

Our approach to enhancing, assessing, and monitoring fidelity [21] was comprehensive and informed by published guidelines [8,14,27,65]. We applied best practice methods for evaluating fidelity to intervention delivery [8,14,27] via independent coding of a random sample of real-world consultations, by trained coders blind to treatment allocation. Unlike much of the behaviour change literature, adherence and competence were evaluated relative to apriori benchmarks and important non-specific effects were explored. Moreover, we provided a detailed analysis of how intervention sessions differed to routine care with regard to intervention components, non-specific effects, and dose. This provides an important context for understanding treatment effects. Although we did not explicitly define proscribed components (i.e., intervention components that are unnecessary and/or unhelpful), our assessment of interpersonal effectiveness” confirms that all sessions were conducted in the appropriate “spirit”.

The design of the current study is such that recruitment was at the level of hospital site. Accordingly, some traditional methods for promoting treatment fidelity (e.g., requiring clinicians to reach a pre-determined competency threshold before involving them in the trial; [8,14,27]) were impractical because the trial utilised existing clinicians employed at participating sites. Therefore, the focus was on early, intensive, real time support together with ongoing monitoring and feedback [21]. This is a key strength in that our team has developed an intervention that was successfully integrated into routine clinical practice. However, as is common in evaluation of complex behaviour change interventions [10] it also means that there is uncertainty about the level of support required to ensure ongoing adherence and competence. Peer supervision models may prove useful (e.g., [66]) and a follow-up pilot study evaluating the utility of peer consultation is underway. Furthermore, we did not formally evaluate dietitian attitudes and beliefs regarding training and the EAT intervention. Findings from a recent systematic review suggest that a positive relationship may exist between clinician attitudes and treatment fidelity [67]. Clinician factors, therefore, represent an important consideration for interpreting fidelity outcomes. Future research would be strengthened by incorporating methodologically sound quantitative (e.g., The Evidence-Based Practice Attitude Scale; [46]) and/or qualitative assessment of clinician attitudes and/or beliefs.

Finally, based on a prominent five factor conceptualisation of fidelity considerations (treatment design, provider training, intervention delivery, treatment receipt, and enactment of treatment skills [8,14,27]), it may appear we have neglected to adequately consider intervention receipt (i.e., the degree to which a patient understands and/or is able to perform intervention skills) and enactment (i.e., the degree to which a patient performs intervention skills within their daily life). For example, client knowledge and/or self-efficacy assessment regarding the importance of maintaining adequate nutrition could have been assessed as an index of treatment receipt. However, as these factors are likely to vary independent of intervention delivery, their utility as an index of fidelity is unclear. Therefore, as previously reported [21] we elected to use the degree to which dietitians delivered the intervention as a proxy for whether or not the intervention was “received” by patients. Regarding enactment, debate remains whether client performance of intervention skills should be defined within the context of fidelity (e.g., [65]). Similarly, in the current trial, if patients were taking steps to maintain adequate nutrition, we expected this to be reflected in improved nutritional status. As nutritional status was the primary outcome of the RCT it is evaluated and reported separately to this fidelity evaluation [17].

## 5. Conclusions

The adherence and competence with which the EAT Intervention was integrated by real-world dietitians into real-world consultations was such that intervention sessions were clearly distinct from the care provided during TAU. This finding is important for developing an intervention that is not only effective, but also “implementable” [8]. Future research efforts should focus on clarifying what level of fidelity is “good enough”, especially within the context of real-world clinical trials. Since intervention delivery was not associated with an increase in session duration or number, it is clearly feasible for use within busy clinical settings. In summary, the EAT Intervention may prove to be a useful model for increasing the routine application of important, but often underutilized evidence based motivational and behavioural strategies by oncology dietitians working with HNC patients.

## Figures and Tables

**Figure 1 nutrients-12-02332-f001:**
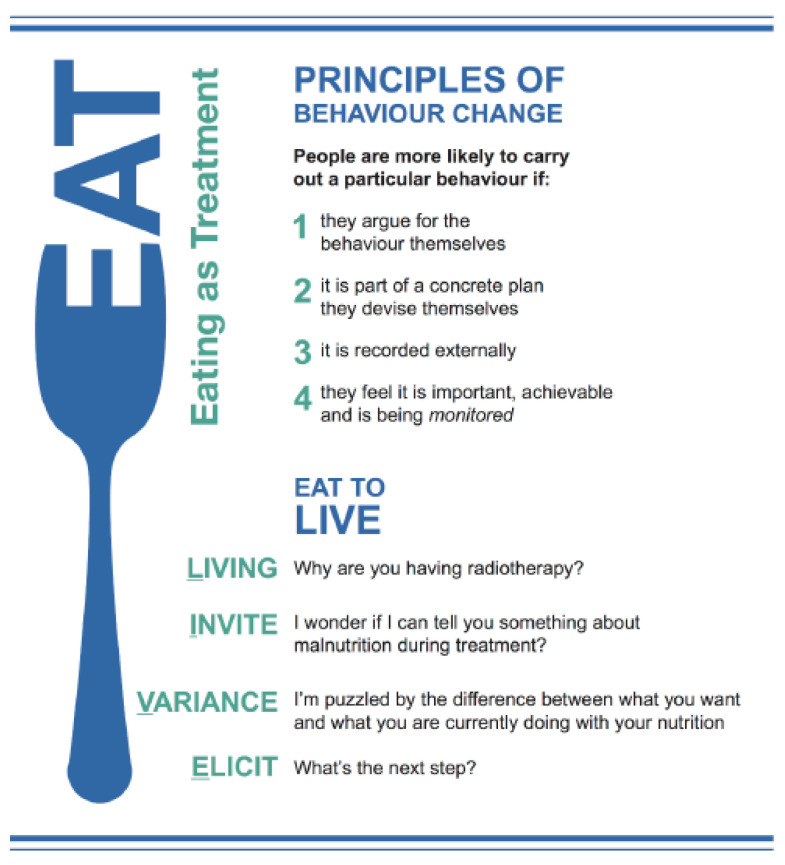
Eating As Treatment (EAT) Intervention: Key principles and prompts for clinicians.

**Figure 2 nutrients-12-02332-f002:**
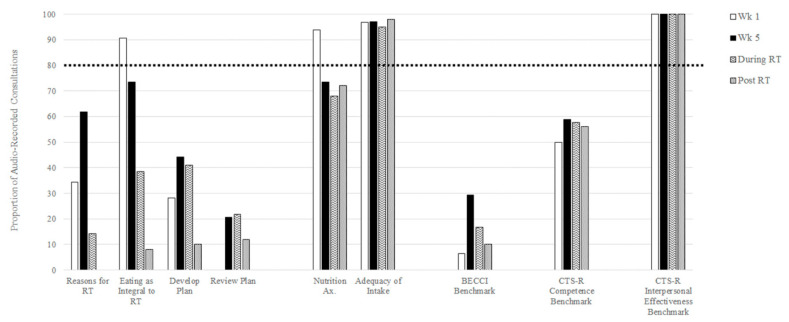
Presence of study specific skills and attainment of fidelity and competence benchmarks within audio-recorded consultations conducted after training in the EAT Intervention.

**Table 1 nutrients-12-02332-t001:** Study Specific Fidelity Checklist.

	YES	NO
Practitioner discusses the adequacy of the patient’s energy intake		
Practitioner conducts a formal/standardised assessment to measure patient nutrition		
Practitioner discusses how eating/nutrition is an integral part of “radiotherapy” treatment		
Practitioner encourages the patient to discuss their reason(s) for undergoing radiotherapy		
Practitioner collaboratively develops a formal, written nutrition plan with the patient		
Practitioner encourages the patient to discuss their progress towards the goals outlined on their written nutrition plan		

**Table 2 nutrients-12-02332-t002:** Differentiation: Impact of training in the EAT Intervention on dietitian use of study specific checklist items ^a^; behaviour change counselling ^b^; attainment of fidelity and competence benchmarks ^a^; dietitian interpersonal effectiveness and competence ^b^; non-specific factors ^b^ and dose ^b^.

	Control (*n* = 196)	Intervention (*n* = 194)	Odds ratio, Beta, or z Score ^†^	Confidence Interval ^†^		*p* ^†^
				Lower	Upper	
ADHERENCE						
Study Specific Checklist						
Reasons for RT	2 (1.2%)	43 (22.1%)	OR = 24.087	3.408	170.247	0.001 **
Eating as Integral to Radiotherapy	49 (25%)	88 (45.3%)	OR = 7.083	2.582	19.431	<0.001 ***
Nutrition Plan	8 (4%)	61 (31.4%)	OR = 55.171	10.875	279.893	<0.001 ***
Review Plan	2 (1.2%)	30 (18.5%)	OR = 4.086	0.446	37.406	0.213
Validated Nutrition Assessment	112 (57.1%)	144 (73.4%)	OR = 0.674	0.299	1.522	0.342
Adequacy of Intake	173 (88.2%)	187 (96.3%)	OR = 0.917	0.211	3.987	0.908
Behaviour Change Counselling Index						
Threshold of 2.57 attained	12 (6.1%)	30 (15.5%)	OR = 11.819	2.617	53.382	0.001 **
Overall Practitioner Score	2.01 (0.39)	2.14 (0.42)	β = 0.315	0.121	0.393	<0.001 ***
“Spirit” of Intervention Delivery						
Meets Dreyfus Threshold (i.e., “Competent”)	196 (100%)	194 (100%)				
Mean CTS-R Interpersonal Effectiveness Score	5.51 (0.73)	5.69 (0.67)	β = −0.023	−0.246	0.180	0.762
COMPETENCE						
CTS-R Application of Behaviour Change Counselling Item						
Meets Dreyfus Threshold (i.e., “Competent”)	69 (35.2%)	109 (56.2%)	OR = 4.176	1.905	9.153	<0.001 ***
Mean CTS-R Application of BCC Competence Score	2.21 (1.08)	2.72 (1.34)	β = 0.386	0.540	1.386	<0.001 ***
NON-SPECIFIC EFFECTS						
Therapeutic Alliance						
Patient Rated	33.01 (4.02)	33.38 (3.07)	z = −0.61	−6.76	3.55	0.542
Dietitian Rated	29.69 (4.60)	31.6 3 (4.34)	z = −0.63	−7.29	3.74	0.528
DOSE						
Number of Dietetic Consultations Attended	10.34 (3.59)	10.05 (2.86)	β = −0.042	−1.026	0.485	0.482
Session Duration (mins)	19:11 (08:11)	19:39 (10:42)	β = 28.55	−85.27	142.37	0.622

Note. BCC = Behaviour Change Counselling; CTS-R = Cognitive Therapy Scale Revised; RT = Radiotherapy ^†^ Adjusted values reported (accounting for hospital site, when during radiotherapy the dietetic consultation was held and calendar time) ^a^ Presented as the number of audio recorded consultations (*n*(%)) coded “yes” (i.e., to demonstrating a given skill/meeting a given benchmark); ^b^ Presented as M (SD); *** *p* < 0.001. ** *p* < 0.01.

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
