# Peer review of "Assessing Adherence, Competence and Differentiation in a Stepped-Wedge Randomised Clinical Trial of a Complex Behaviour Change Intervention"

_nutrients, 2020, doi:10.3390/nu12082332_

Round 1
Reviewer 1 Report
First of all, I would like to thank you for the opportunity to review such a high-quality article. Intervention fidelity is, without a doubt, an issue of great importance. Indeed, problems in the intervention delivery might be one potential explanation for often contradictory results we find in the literature for the efficacy of behaviour change interventions.
The paper explores if the intervention designed to reduce malnutrition in head and neck cancer patients undergoing radiotherapy was adequately delivered by dietitians who participated in the RCT.
Introduction
The topic (and its relevance) is clearly and concisely presented. References support the arguments provided by authors.
Methods
The methods followed in the study are well described. Although, I would appreciate having more information regarding the training provided to the dietitians. There is a reference indicating that more detailed information can be found elsewhere (lines 148 and 149). However, since training constitutes the main change-focused action, including further description would help the reader to better understand and interpret the results.
From the text, rather than an intervention for cancer patients, EAT seems a set of guidelines or recommendations for dietitians aimed at improving their clinical practice. It seems that dietitians were encouraged to use EAT “according to their judgement/clinical needs” (line 140). Later in the discussion, authors suggest that clinician factors, such as preference, might impact the degree to which intervention components might have been implemented (line 363). That raises a question concerning the methodology used during the training: Did training highlight the EAT guidelines as one whole intervention procedure, where all parts were equally relevant? Perceiving the intervention like a set of recommendations might explain why not all components were integrated in the same way.
Results
Nothing to add.
Discussion
Authors conclude that the intervention was successfully integrated. In the Limitation section, they mention as study limitation the fact that they did not consider “intervention receipt and enactment” (line 463). What about dietitians’ attitudes and beliefs regarding training and EAT intervention? Indeed, as authors point out, “the trial utilised existing clinicians employed at participating sites” (line 454). Health practitioners tend to have their own working style that can influence the degree to which changes are implemented. This, again, might explain why not all components were integrated in the same way.
Finally, almost at the end of the paper, authors mention that data regarding the efficacy of EAT intervention have already been published and state that “This finding provides an important foundation for interpreting the improved nutritional status demonstrated by intervention patients” (line 373). Further discussion might help readers to combine results from both articles.
Reviewer 2 Report
The article reports on measures of “how well” a complex behaviour change intervention was implemented in the context of a stepped-wedge cluster trial. Estimates of adherence, competence and differentiation (between treatment and control conditions) are provided. The article presents well-conducted research and can serve as a good exemplar of reporting of crucial secondary trial outcomes. However, some aspects of the work were unclear to me, particularly in the methods, and I believe some changes to the analyses are required.
Major comments
- General: Tables one and two were absent from the review copy and supplementary materials so I was not able to assess these and the results presented there. A further review including these tables is probably required however the comments below will apply to the results in those tables I believe.
- General: The introduction rightly notes that information on compliance and adherence is necessary to properly interpret effectiveness estimates from randomised trials. However, no information is given anywhere about the results from the primary analysis of this intervention and whether it was considered successful or not.
- Introduction: the introduction discusses well the importance of presenting measures of adherence etc. the lack of studies that do this, and the need for better reporting and methods. However, this reads as if this study will be a general methodological study of adherence outcomes rather than the specific assessment of an intervention as little background is provided to the intervention and trial. I think some more information on the context (i.e. malnutrition in head and neck cancer, why the intervention was thought to be needed, and why there are questions of adherence) would be helpful to the reader.
- Methods: there is no information on the nature of the clusters which constitute the basis of the study (the word ‘cluster’ is not mentioned until page 6) nor on the randomisation and duration of the trial. This is important as I believe the training was applied at the cluster level (i.e. to all participating dieticians in each hospital) at the same time, the analyses are stratified by cluster, and the duration of the trial.
- Methods: Could the authors clarify what is meant by week 1 and week 5 in their analyses? I believe it is with reference to weeks one and five of a patient’s course of radiotherapy but it is also referred to as “dietetic interval” (p5 165) to do with “timing of the intervention” (p5 165-6), and “time point during radiotherapy” (p7 254).
- Methods: could the authors define what they mean by “non-specific effects” (p6 233 and elsewhere)? I believe it refers to not being solely measured in the intervention group. However, adherence and competence are discussed both in this section and the one preceding it.
- Methods: The section on “Non-specific effects” is further statistical methods, so may be better incorporated into the preceding section
- Methods: Why is therapeutic alliance the only outcome from the protocol appearing in this paper? Are the other outcomes analysed elsewhere?
- Methods: why does the analysis differ for therapeutic alliance outcome and the other outcomes listed under “Differentiation” (p6)? The analysis proposed for therapeutic alliance seems to be appropriate as it allows for cluster-level correlation, which the differentiation analysis does not.
- Methods: Can the authors clarify the specification of the random effects (p6 243-245)? E.g. “random level individual level”? What is the unit of observation – I assume the patient-dietician consultation? If the audio recordings were randomly sampled from all consultations, I’d assume you’d get many patients with only one observation, so how would an individual-level random effect work here? Similarly with the “random coefficient”, what level does this vary over?
- Methods + results: No statistical tests are mentioned in the methods but “significance” is mentioned in a number of places (e.g. p8 309, p8 313). The use of p-values of statistical significance needs a lot of work if used: what hypotheses are being tested, what is the “significance” cut-off, multiplicity adjustments should be used. Further to that though, the appropriate test statistics should be based on the randomisation scheme since this defines the sampling distribution. And, given the small number of clusters test statistics may also be incorrect. (This also does not seem to have been addressed in the protocol). Some sort of permutation test is probably required. However, this may be beyond the scope of the work presented here. Simpler comparisons between treatment and control groups may be more reliable, such as differences in means, and still serve the purposes of the article.
- Methods: It was unclear to me why the benchmark for the BECCI score was set at 2.57. This was the mean value from the pilot, so assuming the distribution of scores is symmetrical we would only expect 50% of consultations to achieve better than this score, but the threshold is set at 80%. This seems set up to fail.
- Results: p8 298 “link between malnutrition and radiotherapy outcomes”. As far as I can see these outcomes are not described anywhere, nor what this “link” is or how it is estimated.
- Discussion: p11 413-414 “Our findings clearly demonstrate a statistically significant improvement in the application of the EAT Intervention after training.” Can the authors clarify? I thought the training was to provide the EAT intervention itself so it wouldn’t have been provided without it? Perhaps this just requires further information of the elements of the trial.
